# *C. elegans* SAS-1 ensures centriole integrity and ciliary function, and operates with SSNA-1

Keshav Jha[ID], Alexander Woglar¤, Coralie Busso, Georgios N. Hatzopoulos, Tatiana Favez, Pierre Gönczy[ID]*

Swiss Institute for Experimental Cancer Research (ISREC), School of Life Sciences, Swiss Federal Institute of Technology Lausanne (EPFL), Lausanne, Switzerland

¤ Current address: Department of Biochemistry, University of Geneva, Geneva, Switzerland
* pierre.gonczy@epfl.ch

## Abstract

Centrioles are microtubule-based organelles critical for signaling, motility and division. The microtubule-binding protein SAS-1 is homologous to the human ciliopathy component C2CD3 and contributes to centriole integrity in *C. elegans*, but how this function is exerted is incompletely understood. Here, through the generation of a null allele and analysis with U-Ex-STED, we establish that SAS-1 is dispensable for the onset of centriole assembly, but essential for organelle integrity during oogenesis, spermatogenesis and in the early embryo. Additionally, we uncover that SAS-1 is present at the transition zone of sensory neurons, and contributes in a partially redundant manner to ciliary function. Furthermore, we investigate the relationship between SAS-1 and the *C. elegans* Sjögren's Syndrome Nuclear Antigen 1 protein SSNA-1, establishing that SSNA-1 localizes next to the SAS-1 C-terminus in the centriole architecture. Moreover, through molecular epistasis experiments with null alleles of both components, we reveal that SAS-1 is essential for SSNA-1 localization to centrioles during oogenesis and to the transition zone during ciliogenesis. Moreover, using a heterologous human cell assay, we establish that SAS-1 recruits SSNA-1 to microtubules. Overall, our findings help clarify how SAS-1, together with SSNA-1, ensures centriole integrity, and reveal that it contributes to cilium function.

## Author summary

Inside each of our cells lie tiny organelles called centrioles that help organize cell division and form the base of cilia, hair-like projections that sense and respond to the environment. While understanding of how centrioles are built has increased in recent years, less is known about how these organelles maintain their integrity once assembled. In this study, we used the roundworm *C. elegans* to explore the function in centriole integrity of SAS-1, which is related to a human protein linked to a genetic disease that affects cilia and brain development. We discovered

**Data availability statement:** All relevant data are within the manuscript and its Supporting information files.

**Funding:** This work was supported by the Swiss National Science Foundation (grant # 310030_197749 to PG, https://www.snf.ch/en). The funders had no role in study design, data collection and analysis, decision to publish, or preparation of the manuscript.

**Competing interests:** The authors have declared that no competing interests exist.

that centrioles can form in *C. elegans* lacking SAS-1, but that they lose their structural integrity during critical developmental stages. Moreover, we found that SAS-1 contributes to the function of sensory cilia, and that SAS-1 works together with another human-related protein called SSNA-1 to reinforce centriole stability. By uncovering how SAS-1 and SSNA-1 safeguard centriole integrity, our work provides new insights into the mechanisms that ensure proper cell function, and may help explain how defects in similar processes contribute to human diseases.

## Introduction

Centrioles are 9-fold radially symmetric microtubule-based organelles present across the eukaryotic tree of life (reviewed in [1,2]). Centrioles template the axoneme of cilia and flagella, as well as recruit pericentriolar material (PCM) to form centrosomes, which act as microtubule-organizing centers (MTOCs) in animal cells. Through these fundamental roles, centrioles are critical for signaling, motility and division. Cells maintain tight control over centriole number and architecture, and failure to do so can result in disease (reviewed in [3,4]). In most proliferating cells, two pre-existing centrioles seed the formation of one procentriole each in their vicinity. Centrioles are extremely stable in general, but undergo programmed elimination in some circumstances, including during oogenesis of metazoan species (reviewed in [5,6]). Overall, despite their importance, the mechanisms regulating centriole integrity remain incompletely understood.

Systematic analyses in *C. elegans* identified a core set of evolutionarily conserved components necessary for centriole assembly, comprising SAS-7, SPD-2, ZYG-1, SAS-6, SAS-5 and SAS-4 (reviewed in [7,8]). In brief, centriole assembly begins with the kinase ZYG-1 phosphorylating SAS-5 on a platform formed by SAS-7 and SPD-2. Phosphorylated SAS-5 then associates with SAS-6, which can self-assemble into an inner tube element thought to scaffold recruitment of the microtubule-binding protein SAS-4, and then of centriolar microtubules. *C. elegans* centrioles are merely ~150 nm by ~120 nm in dimensions [9–11], yet the distribution of centriolar and PCM components has been determined with high accuracy using Ultra-Expansion coupled to STED super-resolution microscopy (U-Ex-STED) [11].

Although *C. elegans* lacks motile cilia or flagella, centrioles seed the ciliary axoneme of 60 sensory neurons, 56 in the head region and 4 in the tail region of the animal [12,13]. After templating ciliary axoneme formation, centrioles degenerate and are thus absent from mature cilia [14,15]. By contrast, PCM proteins such as SPD-5 and TBG-1 (γ-tubulin) remain at the ciliary base, where they function as acentriolar MTOCs [16,17]. Distal to SPD-5 and TBG-1 lies the transition zone, which marks the beginning of the cilium proper and comprises a central cylinder positioned internal to axonemal microtubules, as well as Y-links that bridge the outside of axonemal microtubules with the ciliary membrane [12,13].

Centrioles are also eliminated in other physiological contexts in *C. elegans*. Thus, centrioles undergo programmed elimination in multiple lineages during embryogenesis, resulting in merely 68 out of 558 cells retaining centrioles by the first larval stage (L1) [18,19]. Moreover, as in other metazoan organisms (reviewed in [5,6]), centrioles are eliminated during *C. elegans* oogenesis, with sperm contributing the sole pair of centrioles to the zygote, thereby ensuring bipolar spindle assembly and faithful chromosome segregation [20,21]. Correlative Light Electron Microscopy (CLEM) analysis revealed that oogenesis centriole elimination begins during meiotic prophase with the loss of the central tube element located inside the wall of centriolar microtubules [21]. Whereas in Drosophila removal of the kinase Polo, and thereby of the PCM, is critical for triggering oogenesis centriole elimination [22], in *C. elegans* the related Polo-like kinases and the PCM do not modulate this process [21].

SAS-1 plays an important role in regulating oogenesis centriole elimination in *C. elegans* [21]. SAS-1 is a C2-domain containing protein [23], which is homologous to human C2CD3, a component essential for assembly of the centriole distal segment, with patient mutations leading to an oral-facial-digital (OFD) syndrome with severe microcephaly and cerebral malformations [24–26]. SAS-1 localizes to the central tube of *C. elegans* centrioles, with the N-terminus located next to centriolar microtubules and the C-terminus positioned more centrally [11]. When expressed in human cells, *C. elegans* SAS-1 binds to and stabilizes microtubules [23]. Importantly, during oogenesis in *C. elegans*, SAS-1 is the first known protein to leave centrioles, which is followed by the loss of organized centriolar microtubules and removal of other centriolar proteins, including SAS-4 and SAS-6 [21]. In the reduction of function allele *sas-1(t1521)*, centrioles are eliminated precociously during oogenesis [21]. Moreover, centrioles contributed to the zygote by *sas-1(t1476)* or *sas-1(t1521)* mutant sperm lose integrity shortly after fertilization [23]. Furthermore, compromising maternal *sas-1* function using these mutant alleles also results in centriole instability, in this case during the course of embryogenesis [23]. Whether a null allele of *sas-1* would result in a more severe phenotype, perhaps complete failure of centriole assembly, remains to be determined.

Here, we generated and analyzed a null allele of *sas-1*, thereby establishing that the protein is dispensable for centriole assembly, but critical for organelle integrity. We also report that SAS-1 localizes to the transition zone of sensory cilia and contributes to their function. Furthermore, we investigated the relationship of SAS-1 with SSNA-1, the worm homologue of Sjögren's Syndrome Nuclear Antigen 1 protein SSNA-1, finding that SAS-1 is essential for recruiting SSNA-1 to centrioles and cilia in *C. elegans*, as well as to microtubules in a heterologous assay. Together, our work helps to understand how SAS-1 and SSNA-1 operate to ensure integrity of the centriole organelle.

## Results

### Loss of SAS-1 impairs centriole integrity during oogenesis

Previous analyses of SAS-1 were conducted with the reduction of function alleles *sas-1(t1476)* and *sas-1(t1521)* [21,23,27]. To investigate the consequences of complete loss of SAS-1, we generated a null allele using CRISPR/Cas9 (S1A Fig). The resulting *sas-1(is13)* mutant harbors a deletion spanning the 5'-UTR and part of exon 1, which removes the ATG start codon and generates a protein null (S1B–S1D Fig). Heterozygous *sas-1(is13)*/hT2 worms give rise to ~20% *sas-1(is13)* homozygous animals at all tested temperatures (S1E Fig). However, such homozygous animals lay very few embryos, which all fail to hatch, indicating that *sas-1* exerts an essential function. As we were particularly interested in analyzing the germline, we considered whether maternally-contributed SAS-1 from heterozygous mothers perdures in the germline of *sas-1(is13)* homozygous mutant progeny. To this end, we performed live imaging of TagRFP-T::SAS-1 in the germline of *sas-1(is13)* homozygous animals derived from *tagRFP-T::sas-1/sas-1(is13)* heterozygous mothers, finding no detectable TagRFP-T::SAS-1 in the female germline or sperm cells of young adults (S1F–S1H Fig).

Equipped with the *sas-1(is13)* null allele, we set out to investigate the consequences of SAS-1 loss, first by immuno-fluorescence on proliferating germ cell nuclei and associated centrioles in the mitotic region of the gonad of young adults (Fig 1B–1D). In control worms, nuclei in this region are similarly sized, with each nucleus associated with centrioles

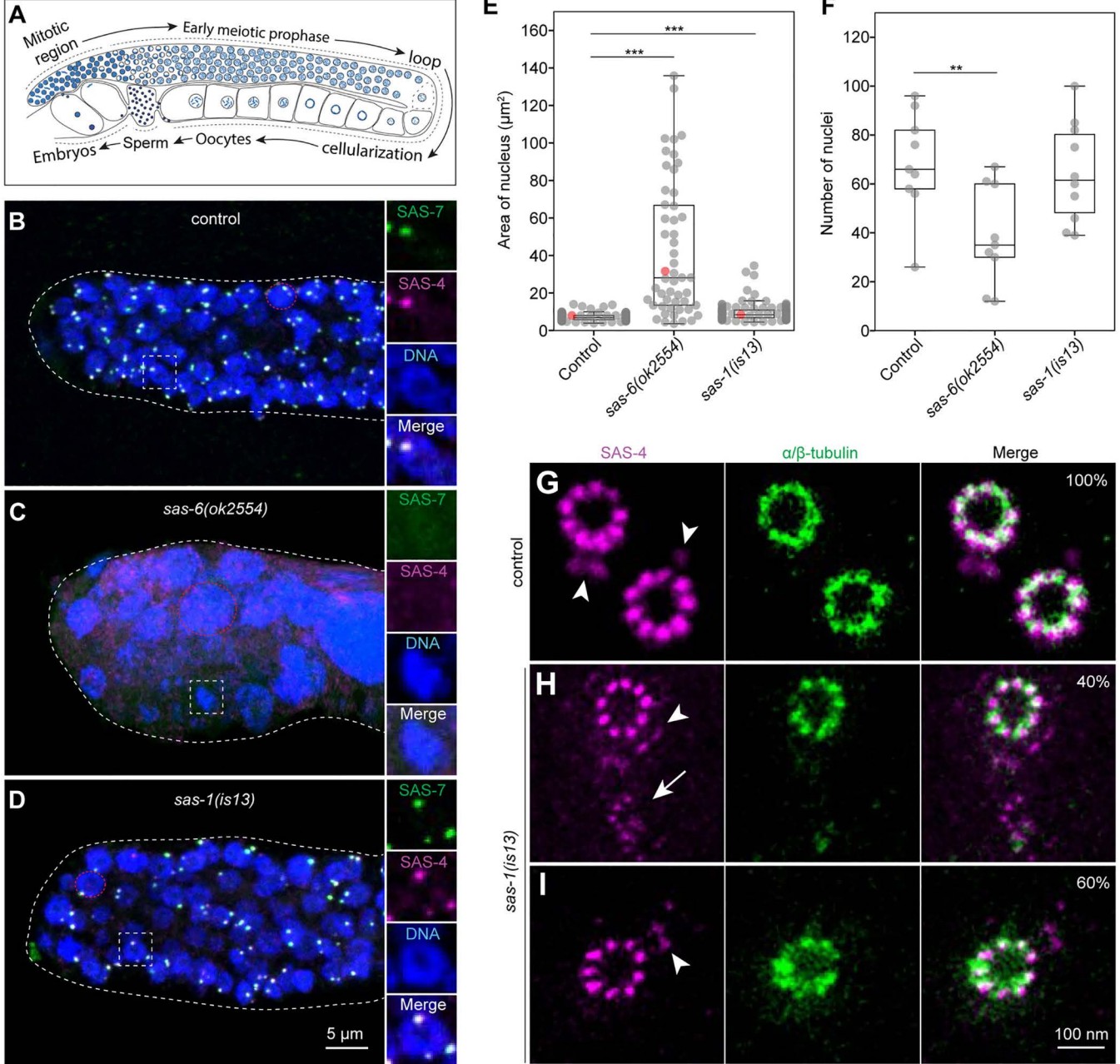

**Fig 1. SAS-1 loss leads to impaired centriole integrity during oogenesis. A.** Schematic of the gonad of a young adult hermaphrodite showing successive stages of gametogenesis, with early mitotic region, followed by early meiotic prophase, before loop region and cellularization. Mature oocytes traverse the spermatheca where fertilization takes places. **B-D.** Representative immunofluorescence images of early mitotic region in fixed gonads of control (B), *sas-6(ok2554)* (C), and *sas-1(is13)* (D) animals expressing GFP::SAS-7, immunostained with antibodies against SAS-4 and GFP. Dotted lines highlight edges of gonad. Square insets on the right are ~1.5 times magnified, whereas dashed red circles exemplify regions measured for nuclear area reported in (E). Brightness and contrast adjusted individually for each image. **E.** Quantification of nuclear area in mitotic region of indicated genotypes. Number of nuclei: N=98 –control, from 5 gonads; 51 –*sas-6(ok2554)*, from 6 gonads; 101 –*sas-1(is13)*, from 5 gonads. Student's two-tailed t-tests, whereby $P<0.001$ (***). **F.** Number of nuclei in mitotic region of gonads of indicated genotypes. Number of gonads=9 –control and *sas-6(ok2554)*; 10 –*sas-1(is13)*. Student's two-tailed t-tests, whereby $P<0.01$ (**). The difference between control and *sas-1(is13)* is not significant ($P=0.68$). **G-I.** Representative U-Ex-STED images of centrioles in the mitotic region of control (G; N=23) and *sas-1(is13)* mutants (H, I; N=15), immunostained for SAS-4 and α/β-tubulin. Arrowheads: procentrioles; arrow: deformed and barely visible centriole. Note that U-Ex-STED might have caused fragile centrioles to further disintegrate as SAS-4::GFP levels in live imaging experiments are unchanged (see S2D–S2F Fig). Scale bar is corrected for the expansion factor of 5.

marked by SAS-7 and SAS-4 (Fig 1B and 1E). By contrast, *sas-6(ok2554)* null mutant animals harbor much fewer nuclei in this region (Fig 1F) [20], which are often much larger than normal (Fig 1E), and almost never associated with SAS-7 or SAS-4 foci (Fig 1C; 3.2%, N = 348), together reflecting failed chromosome segregation and lack of centriole assembly. In contrast to the *sas-6(ok2554)* mutant condition, we found that most nuclei are similarly sized in *sas-1(is13)* null mutants (Fig 1D and 1E), and usually associated with centrioles marked by SAS-7 and SAS-4, as in the control (Fig 1D and 1F; 79.2%, N = 645). We found in addition that the fraction of nuclei lacking detectable centrioles remained essentially constant as germ cell nuclei progressed through oogenesis (S2A–S2C Fig). Furthermore, live imaging of the gonad mitotic region in control and *sas-1(is13)* mutant animals revealed that SAS-4::GFP levels are unchanged compared to the control (S2D–S2F Fig), whereas those of GFP::SAS-7 are lower (S2G–S2I Fig).

We next report high spatial resolution analysis of centriole architecture in the mitotic region of the gonad using U-Ex-STED [11,28]. In the control, the two mature centrioles are recognized in cross-sectional views by the radial distribution of SAS-4 and microtubules (Fig 1G). In addition, SAS-4 is present in the two immature procentrioles at much reduced levels (Fig 1G, arrowheads) [11]. Two types of centriolar configurations were observed in *sas-1(is13)* mutants. In ~40% of cases, one centriole is clearly recognizable, whereas the other one appears deformed and is barely visible (Fig 1H, arrow). In the remaining ~60% of cases, a single centriole is detected instead (Fig 1I). In both configuration types, an immature procentriole is present next to the intact centriole, marked by a weak SAS-4 signal, as in the wild-type (Fig 1H and 1I; arrowheads) [11]. Moreover, we found that the centrioles that are present are slightly narrower than those in the control (S3A and S3B Fig). Furthermore, U-Ex-STED analysis across the mitotic, early prophase, and late prophase regions established that *sas-1(is13)* mutant centrioles remain narrower than in the wild-type, and do not undergo premature elimination (S3A and S3B Fig). We speculate that the centrioles that are deformed or absent all together are the older ones, having been formed in an earlier cell cycle (see Discussion).

Overall, these observations indicate that centrioles can assemble during oogenesis in the absence of SAS-1, but exhibit compromised architecture, including a slightly narrower diameter and selective loss.

## SAS-1 is dispensable for centriole assembly but essential for organelle integrity in the embryo and sperm cells

We sought to test further whether SAS-1 is dispensable for the onset of centriole assembly. To this end, we conducted marked mating experiments, in which otherwise wild-type males expressing TagRFP::SAS-7 were mated with *sas-1(is13)* mutant hermaphrodites expressing SAS-4::GFP (S3C–S3E Fig). If SAS-1 is essential for centriole assembly, then SAS-4::GFP should not be recruited to centrioles in the resulting one-cell stage embryos; moreover, a monopolar spindle is expected to assemble in each blastomere at the two-cell stage, as in embryos lacking maternal *zyg-1* or *sas-5* function [29,30] (S3C–S3E Fig). Contrary to these predictions, however, time-lapse microscopy of such embryos established that two foci of SAS-4::GFP are invariably observed at the one-cell stage (Fig 2A, 0 min; N = 9), and that bipolar spindle assembly always occurs in both blastomeres at the two-cell stage (Fig 2A, 9 min). Interestingly, in addition, we observed that one of the two SAS-4::GFP foci present in each blastomere at the two-cell stage exhibits a weaker signal than the other one initially (Fig 2A, 6 min, insets 3 and 4), and often becomes undetectable by the end of the cell cycle (Fig 2A, 9 min, insets 3 and 4, N = 15/18 centrioles). We reason that the initial presence of the centriole must have sufficed to recruit the PCM, since bipolar spindle assembly systematically takes place in both blastomeres in such two-cell stage embryos (Fig 2A, 9 min; N = 9). Overall, these findings indicate that centriole assembly can occur in the absence of SAS-1, but that the resulting centrioles are unstable and often fail to persist through subsequent divisions.

We analyzed also the impact of complete loss of SAS-1 in sperm cells. In previous work, electron-microscopy (EM) analysis did not reveal an apparent difference between wild-type and *sas-1(t1476)* sperm centrioles, although mutant sperm centrioles were then rapidly eliminated in the embryo [23]. Here, we used U-Ex-STED to examine the molecular architecture of mature sperm centrioles before fertilization. In the control, both SAS-4 and microtubules localize to the two sperm centrioles (Fig 2B). By contrast, *sas-1(is13)* mutant sperm centrioles exhibit two types of configurations, both

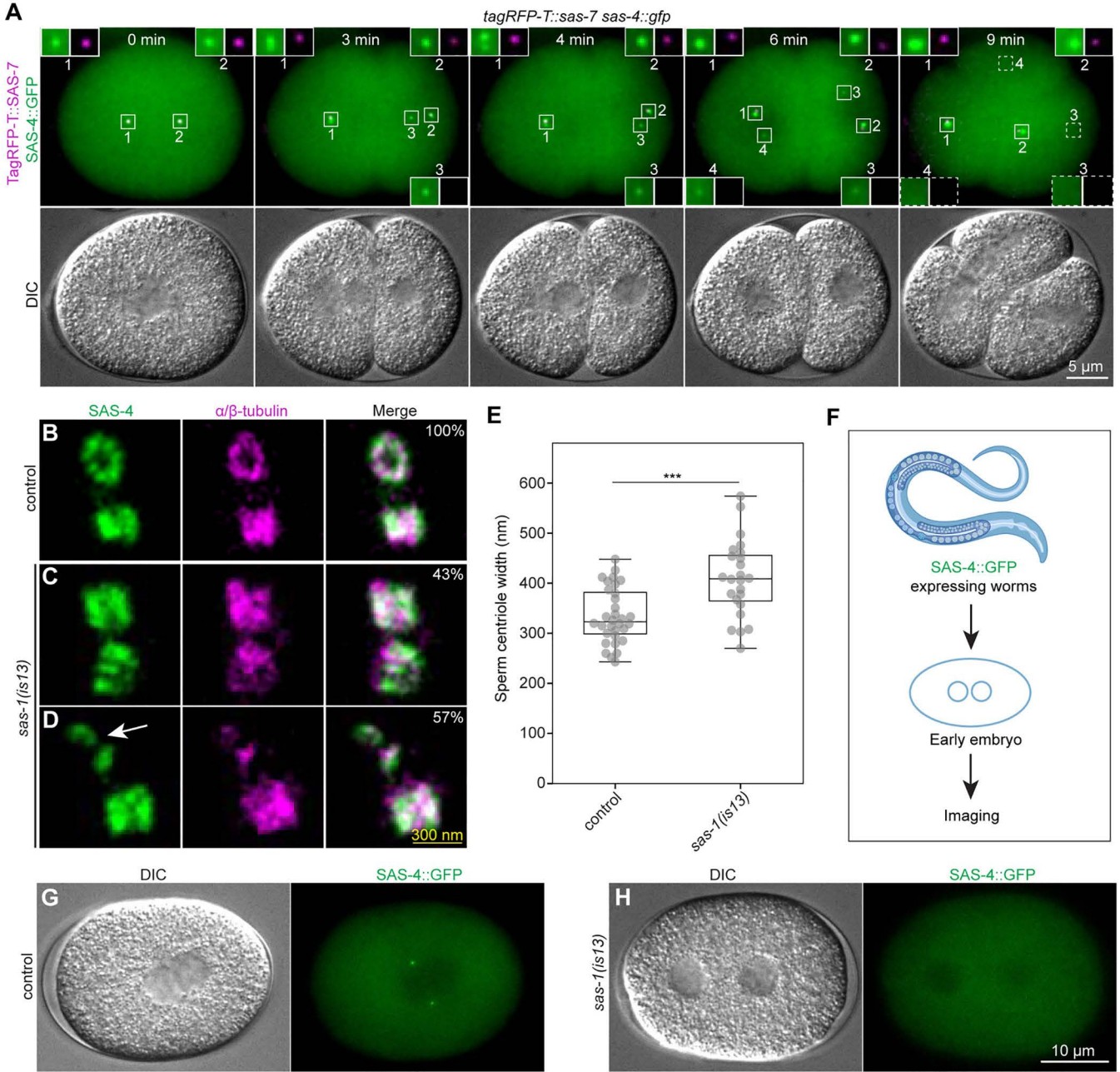

**Fig 2. SAS-1 is dispensable for centriole assembly but essential for organelle integrity. A.** Snapshots from live imaging of early embryo stemming from mating males expressing TagRFP-T::SAS-7 with *sas-1(is13)* mutant hermaphrodite expressing SAS-4::GFP (see S3C–S3E Fig). TagRFP-T::SAS-7 marks sperm-contributed centrioles (insets 1 and 2), whereas SAS-4::GFP marks all centrioles assembled after fertilization. T = 0 corresponds to meta-phase in the one-cell stage. Insets 3 and 4 mark degenerating centrioles assembled next to sperm-contributed centrioles. The DIC image is a single plane, whereas the two merged fluorescent channels are maximum intensity projection of selected Z-planes. **B-D.** Representative U-Ex-STED images of sperm centrioles in control (B; N = 16) and *sas-1(is13)* mutants (C, D; N = 23), immunostained for SAS-4 and α/β-tubulin. In *sas-1(is13)* mutants, two configuration types were observed: (C) both centrioles are present but widened, and (D) one centriole is present and widened, whilst the other one is ruptured (arrow). The percentages in the top right indicate the frequency of each type. Note that U-Ex-STED of sperm centrioles requires methanol fixation (unlike that of gonad centrioles in female germline shown in Fig 1G–1I), compromising resolution. Moreover, we found that methanol fixed samples do not expand linearly with expansion gels; as a result, the scale bar in this case has not been corrected for the expansion factor and is shown in yellow to highlight this fact. **E.** Quantification of sperm centriole width (not corrected for expansion factor), excluding centriole fragments, from (B-D). N = 32 -control, 24 -*sas-1(is13)*). Student's two-tailed t-tests, whereby $P < 0.001$ (***). **F.** Schematic of live imaging of one-cell stage embryos in G and

H (created and adapted from BioRender (https://BioRender.com/s7jx4jh)https://BioRender.com/s7jx4jh). **G, H.** Snapshots from live imaging at the time of pronuclear meeting in control (G; N = 10) and *sas-1(is13)* mutant (H; N = 12) one-cell stage embryos. Images were acquired every minute. The DIC channel is a single plane, whereas the SAS-4::GFP channel is a maximum intensity projection of selected Z-planes.

distinct from the control. In ~43% of cases, both centrioles are detectable and exhibit an increased width compared to the control (Fig 2C). In the remaining ~57%, one centriole is present and wider than in the control, whereas the other centriole is ruptured (Fig 2D, white arrow). By analogy to the situation in the germline mitotic region, we propose that the ruptured centriole is the oldest and therefore may have endured greater mechanical stress during the two meiotic divisions (see Discussion).

What is the fate of *sas-1(is13)* mutant sperm centrioles following fertilization? To address this question, we used live imaging to monitor sperm-contributed centrioles marked by SAS-4::GFP (Fig 2F). In the control, both sperm-contributed centrioles are invariably visible at pronuclear meeting (N = 10) (Fig 2G). By contrast, we found that a SAS-4::GFP focus is never present at the equivalent stage in *sas-1(is13)* mutant embryos (N = 12), although a faint focus could be detected sometimes earlier in the cell cycle (N = 8/12) (Fig 2H). Similar results were obtained when sperm centrioles were monitored using GFP::SAS-7 (S3F and S3G Fig). Together, these observations indicate that the loss of SAS-1 allows sperm centrioles to be assembled, but also compromises their integrity, presumably leading to their rapid demise in the embryo following fertilization.

## SAS-1 kinetics during embryogenesis

We set out to investigate SAS-1 distribution during programmed centriole elimination in embryogenesis [19]. Embryos complete their proliferative phase by ~350 minutes post-fertilization (mpf) [31], by which time most of the 558 cells present at hatching have been generated. In the subsequent morphogenesis stage, embryos sequentially adopt characteristic bean, comma, 1.5-fold, and 2-fold configurations, accompanied by extensive lineage-specific centriole elimination [19]. We performed lattice light sheet live imaging of embryos expressing TagRFP-T::SAS-7 and GFP::SAS-1 from the bean stage (~360 mpf) through the 2-fold stage (~460 mpf) (Fig 3A and S1 Movie), after which twitching prevents faithful centriole monitoring. We found that the number of GFP::SAS-1 foci decreases from ~550 (± 28) at the bean stage to ~474 (± 43) at the 2-fold stage (Fig 3B), a number higher than the ~220 (± 50) determined previously at this stage for GFP::SAS-7 [19]. Accordingly, some GFP::SAS-1 foci that do not appear to bear TagRFP-T::SAS-7 are present at the 2-fold stage, in particular in the embryo anterior, although this may reflect in part weaker signal intensity for TagRFP-T::SAS-7 (Fig 3A, 2-fold stage, inset). Moreover, whether these GFP::SAS-1 positive foci correspond to *bona fide* centrioles or merely to a remaining focus of SAS-1 protein remains to be determined.

Regardless, despite this difference between the two markers at the 2-fold stage, by the end of embryogenesis, TagRFP-T::SAS-1 is present typically in L1 larvae in those cells that also harbor foci of GFP::SAS-7, with a few additional foci positive for only one of the two fusion proteins, usually at low signal intensity, suggestive of spurious labelling (Fig 3C and 3D). In addition to this general colocalization, we observed strong bilateral TagRFP-T::SAS-1 foci in both anterior and posterior of the animal that do not colocalize with GFP::SAS-7 (Fig 3C, arrows). Based on their position, these extra TagRFP-T::SAS-1 foci appear to mark the base of anterior and posterior sensory ciliated neurons.

## SAS-1 localizes at the transition zone of sensory cilia

We set out to address whether SAS-1 truly localizes to the ciliary base and, if so, where exactly compared to previously mapped proteins. Live imaging of L1 larvae revealed a consistent separation between TagRFP-T::SAS-1 and foci of the ciliary base protein SPD-5, with TagRFP-T::SAS-1 foci being positioned slightly more anterior to GFP::SPD-5 foci in anterior cilia (Fig 4A), and slightly more posterior to GFP::SPD-5 foci in posterior cilia (Fig 4B). These patterns suggest that SAS-1 does not localize at the ciliary base proper, but instead in the transition zone. Accordingly, we found that TagRFP-T::SAS-1 colocalizes with the transition zone protein MKSR-2 [32], both in anterior and posterior cilia (Fig 4C and

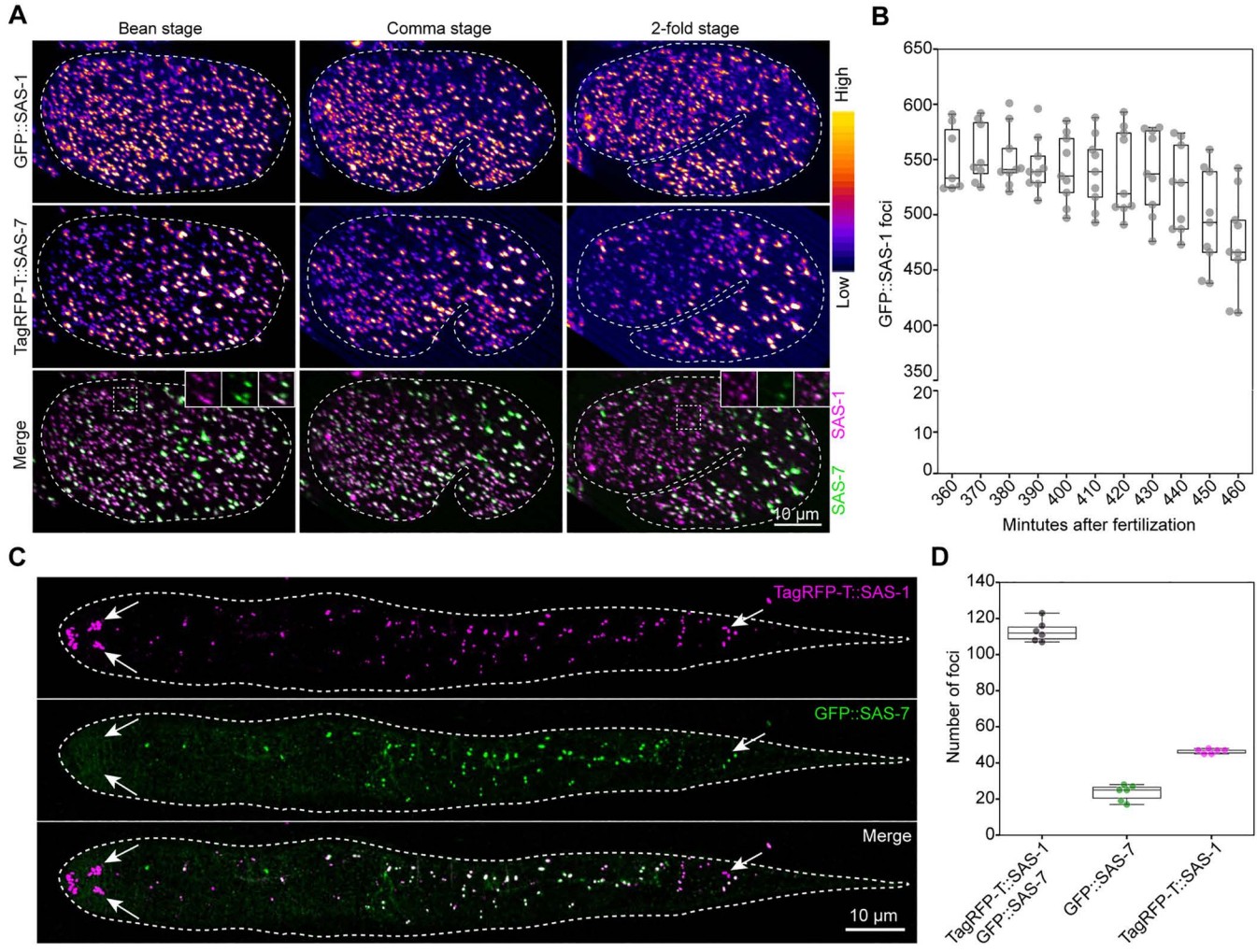

**Fig 3. SAS-1 departs from centrioles during embryogenesis programmed centriole elimination.** A. Snapshots of time-lapse of embryos expressing TagRFP-T::SAS-7 and GFP::SAS-1 at bean, comma, and 2-fold stages, as indicated. Top and middle rows show single channels displayed with a "fire" LUT, the bottom row the merge of the two channels, with insets highlighting foci in the embryo anterior. See also S1 Movie. B. Quantification of number of foci positive for TagRFP-T::SAS-1 from the bean stage until the 2-fold stage, 10 min before twitching (N = 9). C. Live imaging of L1 larva expressing TagRFP-T::SAS-1 and GFP::SAS-7. Anterior is to the left. Arrows point to foci strongly positive solely for TagRFP-T::SAS-1 in the anterior and posterior of the animal. F. Quantification of number of foci positive for TagRFP-T::SAS-1 and GFP::SAS-7, or for just one of the two fusion proteins (N = 6 L1 larvae).

4D). We set out to investigate whether SAS-1 is present within the confines of axonemal microtubules, where the central cylinder is located, or instead external from them, where Y-links are located. To this end, we placed L1-L2 larvae expressing TagRFP-T::SAS-1 in a solution containing the microtubule probe SPY650-tubulin, reasoning that this small molecule could be taken up by sensory cilia. As shown in Fig 4E, we found this to be the case indeed. AiryScan imaging indicates that TagRFP-T::SAS-1 is present within the confines of the axonemal microtubule signal marked by SPY650-tubulin (Fig 4F), suggesting that SAS-1 resides in the central cylinder of the transition zone [33]. This localization echoes that of SAS-1 at the central tube inside the centriolar microtubule wall [11].

We set out to test whether SAS-1 exerts a function at sensory cilia. We found that the TagRFP-T::SAS-1 signal at the transition zone declines during the larval stages (S4A Fig), suggesting that a putative function would be exerted during

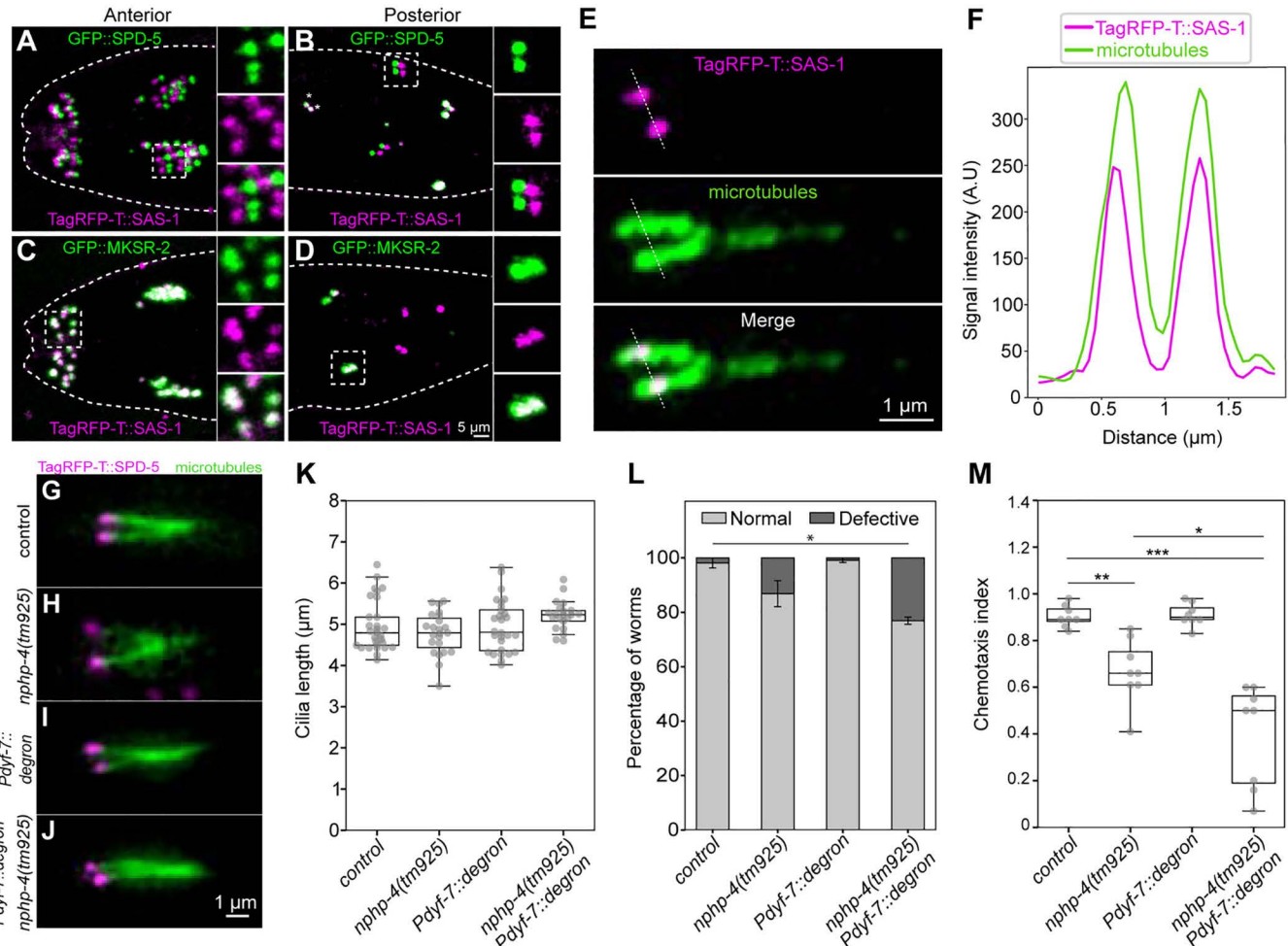

**Fig 4. SAS-1 is a component of the transition zone and contributes to ciliary function. A-D.** Live imaging of anterior (amphid; A, C) and posterior (phasmid; B, D) cilia of L1 larvae expressing GFP::SPD-5 and TagRFP-T::SAS-1 (A, B) or GFP::MKSR-2 with TagRFP-T::SAS-1 (C, D). Insets are magnified approximately two-fold. **E.** Live Airyscan imaging of posterior cilium of L1 larva expressing TagRFP-T::SAS-1, with axonemal microtubules stained with SPY650-tubulin. **F.** Line profile of signal intensity of TagRFP-T::SAS-1 and SPY650-tubulin along the dotted line in Fig 4E. **G-J.** Representative images of posterior cilia in L4 larvae, stained with SPY650-tubulin (green) marking axonemal microtubules and TagRFP-T::SPD-5 (magenta) marking the ciliary base in control (G), *nphp-4(tm925)* (H), *Pdyf-7::degron* (I), and *Pdyf-7::degron nphp-4(tm925)* (J). **K.** Quantification of ciliary length in Fig 4G–4J. Segmented lines were drawn starting from the TagRFP-T::SPD-5 foci and tracing the SPY650-tubulin signal to quantify ciliary length. Cilia quantified: N = 29 -control, from 15 animals; N = 23 -*nphp-4(tm925)* from 14 animals; N = 28 -*Pdyf-7::degron*, from 14 animals, N = 21 -*Pdyf-7::degron nphp-4(tm925)*, from 10 animals. Student's two-tailed t-tests, which were all not significant compared to control vs *nphp-4(tm925)*: $P = 0.24$, *Pdyf-7::degron*: $P = 0.76$, *Pdyf-7::degron nphp-4(tm925)*, $P = 0.15$. **L.** Dye-filling assay of indicated strains. Dye-filling was scored as normal if all four phasmid neurons were stained with DiI, and defective otherwise. The assay was conducted twice. Student's two-tailed t-tests and comparison were made with the percentage of normal dye-filling in the control vs *nphp-4(tm925)*: $P = 0.22$, *Pdyf-7::degron*: $P = 0.69$, *Pdyf-7::degron nphp-4(tm925)*, $P = 0.015$ (*). **M.** Chemotaxis index for animals of indicated genotypes. Data points are from eight experiments. Student's two-tailed t-tests, whereby $P < 0.05$ (*); $P < 0.01$ (**); $P < 0.001$ (***).

early ciliogenesis. To remove SAS-1 from developing cilia, given that *sas-1(is13)* mutant embryos fail to develop, we deployed ZIF-1-mediated degradation of endogenous SAS-1 using the neuron-specific *dyf-7* promoter, which expresses during early ciliogenesis (S4B Fig) [34,35]. We found that ZIF-1-mediated depletion of SAS-1 did not reduce TagRFP-T::SPD-5 levels at the ciliary base (S4C, S4E and S4G Fig). Because transition zone proteins can act redundantly [33,36,37], we also tested whether TagRFP-T::SPD-5 distribution upon SAS-1 removal is compounded by loss of the Y-link

protein NPHP-4 using the null allele *nphp-4(tm925)* [38,39], but found this not to be the case (S4C–S4G Fig). Similarly, the levels of the axonemal motor protein CHE-11::mKate2 are not reduced by SAS-1 depletion alone, nor are they further diminished by SAS-1 removal in the *nphp-4(tm925)* mutant background (S4H–S4L Fig). Moreover, measurements of cilia length in L4 larvae using SPY650-tubulin did not reveal significant differences in either single or double depletion conditions (Fig 4G–4K).

To begin assessing ciliary function, we performed dye-filling assays in phasmid neurons. We found that whereas SAS-1 removal alone or *nphp-4(tm925)* alone has no effect, SAS-1 removal in the *nphp-4(tm925)* mutant background results in a modest but significant dye-filling defect (Fig 4L). Dye-filling also enabled measurement of dendrite length, which was unchanged across all conditions (S5A–S5E Fig). We also conducted a chemosensation assay to determine whether this aspect of ciliary function is perturbed upon SAS-1 removal. In this assay, worms are placed in the center of a plate that contains a drop of the attractant Isoamyl alcohol on one side and a drop of the repellent Ethanol on the other (S5F Fig) [40]. Wild-type animals move towards the attractant, whereas animals with defective chemosensation fail to do so, a preference quantified with a chemotaxis index. As shown in Fig 4M, we found that ZIF-1-mediated depletion of SAS-1 from cilia alone does not result in a detectable chemosensation defect, but significantly enhances the chemosensation defect of *nphp-4(tm925)* mutant animals. Taken together, these findings establish SAS-1 as a transition zone protein that contributes in a partially redundant manner to proper ciliary function in *C. elegans*.

## SSNA-1 colocalizes with SAS-1 at the central tube of centrioles

Prompted by the apparent phenotypic similarity between embryos depleted of maternal SAS-1 [23] and those lacking the Sjögren's Syndrome Nuclear Antigen 1 (SSNA-1) protein [41], we set out to investigate a possible relationship between these two components. Live imaging of control (Fig 5A) and *ssna-1(bs182)* null mutant (Fig 5B) embryos confirmed the frequent occurrence of multipolar spindles in two-cell stage *ssna-1(bs182)* embryos (6/14) [41]. Moreover, we generated *sas-1(t1476) ssna-1(bs182)* double-mutant embryos and found that, in contrast to either single mutant, sperm-contributed GFP::SAS-7 foci are not detected at pronuclear meeting (S6A–S6D Fig), as is the case in single *sas-1(is13)* null mutant (see S3G Fig).

SSNA-1 has been reported to colocalize with SAS-1 inside the microtubule wall of centrioles [41]. We set out to investigate the localization of SSNA-1 with respect to that of SAS-1 using U-Ex-STED. We analyzed centrioles in early meiotic prophase because the relatively thin and well-spread gonad tissue permits improved spatial resolution, which is difficult to achieve in early embryos due to tissue thickness and optical limitations. The N-terminus of SAS-1 localizes next to centriolar microtubules and SAS-4 (Fig 5C), whereas the C-terminus maps to the more inward central tube (Fig 5D) [11]. Using the diameter of the SAS-4 signal as a ruler in these experiments, we found that SSNA-1::SPOT localizes close to the C-terminus of SAS-1 (Fig 5D). Interestingly, AlphaFold2 multimer predictions [42] indicate that SAS-1 interacts with SSNA-1 through its C-terminal alpha-helix (amino acids 546–570) (S6E and S6F Fig), as reported also elsewhere [41]. Overall, these findings suggest that SAS-1 and SSNA-1 act together to maintain centriole integrity.

## SAS-1 is required for SSNA-1 recruitment to centriole, cilia and microtubules

Given their close localization at centrioles and potential joint function, we addressed whether SSNA-1 departs from centrioles concomitantly with SAS-1 during oogenesis. As shown in Fig 5E–5H, we found that SSNA-1 leaves centrioles just after SAS-1, and before other centriolar components like SAS-4, in line with observations by others [41]. Next, we investigated the epistatic relationship between SAS-1 and SSNA-1 during oogenesis. We found that TagRFP-T::SAS-1 departs precociously from centrioles in *ssna-1(bs182)* mutants (Fig 6A–6C). Strikingly, in addition, SSNA-1::SPOT is absent from centrioles in *sas-1(is13)* mutants (Fig 6D). SSNA-1 also localizes at sensory cilia, likely at the transition zone with SAS-1 (S7C Fig). We found a similar relationship between SAS-1 and SSNA-1 in cilia as during oogenesis, with SAS-1 levels being decreased in the absence of SSNA-1 (S7A and S7B Fig), and SSNA-1 failing to localize upon neuron-specific

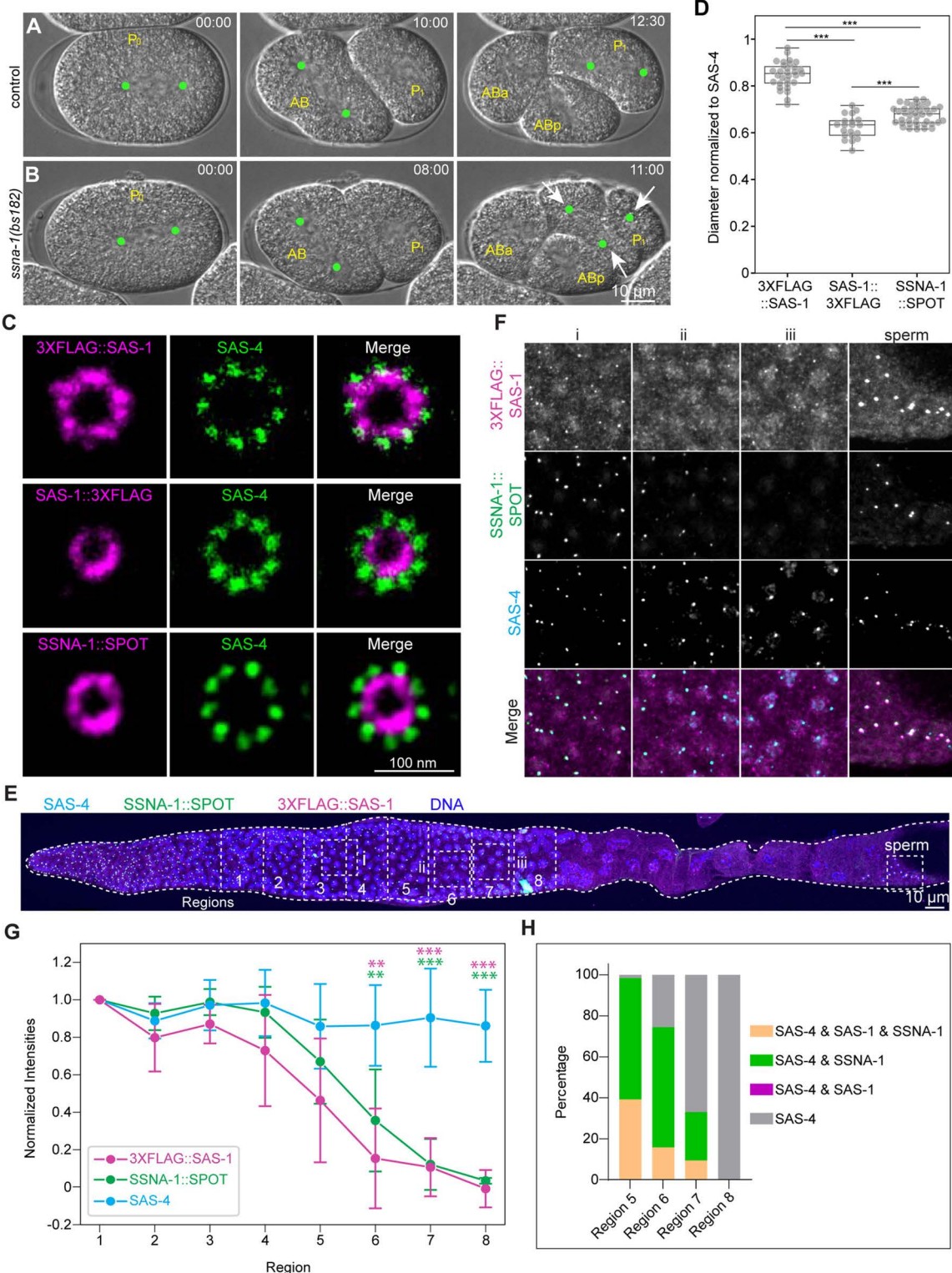

**Fig 5. SSNA-1 is part of the central tube, together with the SAS-1 C-terminus. A, B.** Snapshots from DIC live imaging of early embryos of control (A; N = 8) and *ssna-1(bs182)* embryo (B; N = 14). Time indicated in min:sec; images acquired every 30 sec. Cell designations are shown in yellow; green dots indicate spindle poles, arrows the poles of a tripolar spindle. **C.** Representative U-Ex-STED of meiotic prophase centrioles of animals expressing

FLAG-tagged SAS-1 (on either N- or C-terminus, as indicated), and SSNA-1::SPOT, immunostained together with SAS-4. Scale bar normalized to expansion factor of 5. **D.** Quantification of centriole diameter of 3XFLAG::SAS-1 (N = 31), SAS-1::3XFLAG (N = 20) and SSNA-1::SPOT (N = 40) distributions normalized with SAS-4 ring diameter. Student's two-tailed t-tests, whereby $P < 0.001$ (***). **E.** Representative immunostaining of fixed gonads with antibodies against SAS-4 and FLAG, as well as SPOT. Different sections of the gonad stitched together for visualization. The meiotic prophase area was divided into eight regions, going backward from cellularization of meiotic nuclei (region 8). Numbered regions indicate where the quantification reported in G and H has been conducted. Dashed squares (i, ii, iii) indicate areas magnified in **F. F.** About three times magnified insets from (E) across meiotic prophase. **G.** Normalized signal intensities for SAS-4, 3XFLAG::SAS-1, and SSNA-1::SPOT across eight regions indicated in (E). Background subtracted signal intensities for each region were averaged and then normalized with the mean intensity of region 1. Mean ± SD for each region (N = 6 gonads, with 15 – 25 foci per region for each gonad). Student's two-tailed t-tests were performed for each region comparing signal intensities of SAS-4 with 3XFLAG::SAS-1 (magenta stars) and with SSNA-1::SPOT (green stars); $P < 0.01$ (**); $P < 0.001$ (***). **H.** Quantification of foci comprising SAS-1, SSNA-1 and SAS-4 in the indicated combination in the last four regions of (E) (N = 6 gonads; total number of foci analyzed: 197 for region 5, 133 for region 6, 106 for region 7, 74 for region 8).

SAS-1 depletion (Fig 6E). Overall, we conclude that SAS-1 is essential for SSNA-1 localization at centrioles and at the ciliary base, whereas SSNA-1 reciprocally maintains SAS-1.

To further elucidate the relationship between SAS-1 and SSNA-1, as well as to clarify their interaction with microtubules, we expressed the worm proteins in human U-2 OS cells (U2OS hereafter) (Fig 6F–6M). Consistent with previous work [23], we observed that wild-type SAS-1-GFP (Fig 6G), but not a SAS-1-P419S-GFP protein corresponding to the *sas-1(t1476)* mutant (Fig 6H), localizes with microtubules and induces their bundling. By contrast, SSNA-1-Myc forms large cytoplasmic aggregates that do not colocalize with microtubules (Fig 6J). Importantly, SAS-1-GFP co-expression recruits SSNA-1-Myc to microtubules (Fig 6K), whereas this colocalization is not observed when SSNA-1-Myc is co-expressed with SAS-1-P419S-GFP (Fig 6L). Therefore, SSNA-1 localization on microtubules depends on interaction with functional SAS-1 in this assay. We further tested whether a SAS-1 variant lacking the small C-terminal region predicted by AlphaFold2 to interact with SSNA-1 (SAS-1ΔC, lacking amino acids 546–570) could recruit SSNA-1 to microtubules. Strikingly, although SAS-1ΔC localizes to microtubules similarly to full-length SAS-1 (Fig 6I), it fails to recruit SSNA-1 (Fig 6M), further demonstrating that this short C-terminal alpha-helical segment is critical for the interaction between SAS-1 and SSNA-1.

## Discussion

The mechanisms through which the centriole organelle maintains integrity once assembled are only beginning to be understood. Here, we analyzed how SAS-1 contributes to this process in *C. elegans*. Through the generation of a null allele and tissue-specific removal, we demonstrate that SAS-1 is dispensable for assembly of the centriole, but required for maintaining its integrity during oogenesis, spermatogenesis and embryogenesis. Moreover, we reveal that SAS-1 is present at the transition zone of sensory cilia and contributes to cilium function. Furthermore, our findings indicate that the ability of SAS-1 to bind microtubules enables recruitment of SSNA-1 to centrioles and cilia.

### SAS-1 is dispensable for centriole assembly but essential for centriole integrity

Centrioles are exceptionally stable organelles that are resistant to microtubule depolymerization induced by drugs or low temperature. Reflecting this unusual stability, several structural centriolar proteins exhibit at most marginal turnover once incorporated into the organelle. Thus, pioneering experiments in vertebrate cells established that centriolarα/β-tubulin do not exhibit substantial exchange during an entire cell cycle [43]. Likewise, in *C. elegans*, centriolar β-tubulin, SAS-6 and SAS-4 incorporated in sperm centrioles undergo little to no exchange over many cell cycles in the ensuing embryo [44]. What then triggers programmed disassembly of such a stable organelle?

Previous analysis conducted with two reduction of function alleles, *sas-1(t1476)* and *sas-1(t1521)*, indicated that SAS-1 is key for centriole integrity in *C. elegans* [21,23]. However, because these are not null alleles, these early findings left open the possibility that SAS-1 is also required for centriole assembly. To address this possibility, we generated the null

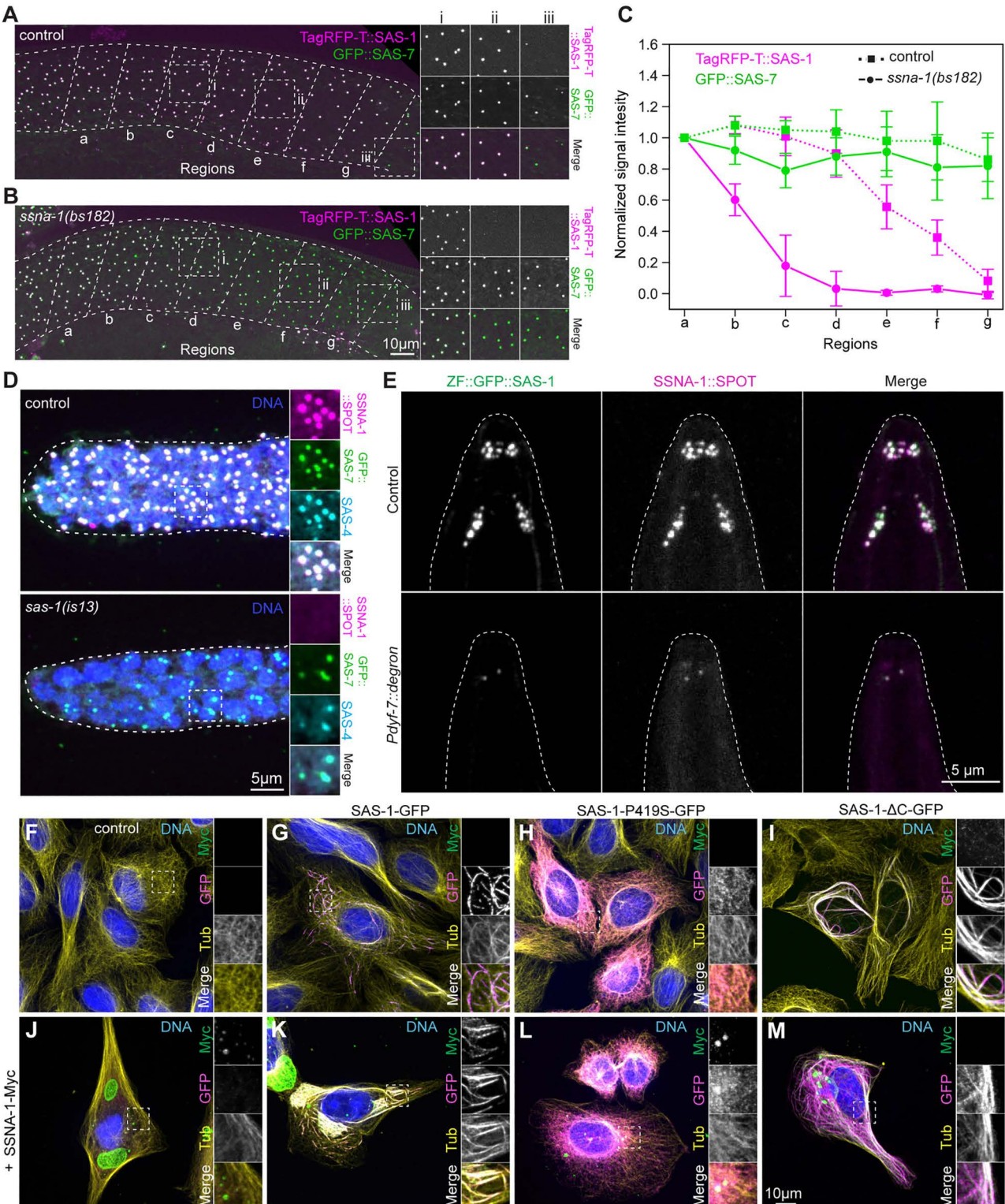

**Fig 6. SAS-1 recruits SSNA-1 on centrioles and cilia. A, B.** Live imaging of gonads in worms expressing GFP::SAS-7 and TagRFP-T::SAS-1 in control (A) and *ssna-1(bs182)* mutant (B). Dashed squares (i, ii, iii) indicate areas magnified approximately 1.5 times on the right. Gonads were divided into seven regions going backward from the loop (g). **C.** Quantification of signal intensities for GFP::SAS-7 (green) and TagRFP-T:::SAS-1 (magenta) across

regions indicated in (A) and (B). Background subtracted signal intensities for each region were averaged and then normalized with the mean intensity of region a. Mean ± SD for each region; N = 7 -control and 9 -ssna-1(bs182) gonads, with typically 5-10 foci quantified per region. **D.** Immunostaining of control (N = 6) and *sas-1(is13)* mutant (N = 8) gonads stained for SAS-4, GFP and SSNA-1::SPOT. Dashed squares indicate areas in the mitotic region of the gonad magnified approximately 1.5 times on the right. **E.** Representative immunofluorescence images of control L1 larva (N = 10), as well as L1 larva following Pdyf-7-driven ZIF-1-mediated SAS-1 depletion from ciliated neurons (N = 11), immunostained for SSNA-1::SPOT. **F-M.** Immunostaining of control U2OS cells (F, no transfection) or U2OS cells expressing SAS-1-GFP (G), SAS-1-P419S-GFP (H), SAS-1-ΔC-GFP (I), SSNA-1-Myc (J), SAS-1-GFP and SSNA-1-Myc (K), SAS-1-P419S-GFP and SSNA-1-Myc (L), SAS-1-ΔC-GFP and SSNA-1-Myc (M), immunostained for GFP (magenta), α-tubulin (yellow) and Myc (green). Boxes indicate location of insets, which are magnified approximately 2-fold.

allele *sas-1(is13)*. We found that germ line nuclei in the mitotic region of *sas-1(is13)* mutants possess centrioles with accompanying procentrioles, indicating that centrioles can assemble in the absence of SAS-1. Analysis of these germ line nuclei shed further light on the fate of centrioles in the absence of SAS-1 (S8A and S8B Fig). Thus, U-Ex-STED uncovered that the first centriole in each pair appears intact, but is slightly narrower, whereas the second centriole is either severely damaged or else not detectable. We propose that this second centriole is the oldest one in each pair, which is known to serve as the principal site of PCM recruitment during mitosis [11]. As a result, this older centriole may suffer damage from forces acting on spindle poles during mitotic division. By contrast, the procentriole matures into a full-fledged centriole only by the end of mitosis [11], such that it would less be subjected to damage and remain comparatively intact until the next mitotic division (S8A and S8B Fig). Furthermore, we hypothesize that those configurations in which the second centriole is not detectable correspond to a centriole that formed at least one cell cycle before the second centriole in these configurations where it is merely damaged (S8B Fig). Once germ line nuclei enter the meiotic cell cycle, centrioles are no longer subject to mitotic forces, potentially explaining why this characteristic pattern remains in place until the onset of programmed elimination in diplotene.

What could explain the fact that centrioles lacking SAS-1 are slightly narrower than in the wild-type? Intriguingly, we observed that GFP::SAS-7 is decreased in *sas-1(is13)* mutant centrioles in the germline. Since *sas-7* mutations cause loss of paddlewheel structure and reduction of centriole diameter [9], decreased GFP::SAS-7 might explain the narrower diameter of *sas-1(is13)* mutant centrioles. It will be interesting to investigate whether proteins that localizes to the paddlewheel, including HYLS-1, SPD-5, and PCMD-1 [11], are likewise affected in the absence of SAS-1.

Analysis of sperm centrioles by U-Ex-STED is also compatible with the lack of SAS-1 resulting in a gradual loss of centriole integrity. Here, U-Ex-STED uncovered that the first centriole in each pair appears intact, but slightly larger, whereas the second centriole is either ruptured or else seemingly normal. As for the female germ line, these configurations could reflect centriole age (S8C and S8D Fig). By analogy and extension, we propose that the ruptured centriole corresponds to the older one, which was present at the onset of the first meiotic division, with those made thereafter being intact, albeit larger than in the wild-type, perhaps reflecting forces stretching them during the second meiotic division. Regardless, all sperm centrioles formed in the absence of SAS-1 are fragile, since they begin to recruit PCM and participate in mitotic spindle assembly in the ensuing zygote, but are progressively lost during the course of the first cell cycle. Together, these findings establish that SAS-1 is dispensable for centriole assembly but essential for maintaining centriole integrity.

Overall, we conclude that the function of SAS-1 is distinct from that of the centriolar assembly factors SAS-7, SPD-2, ZYG-1, SAS-4, SAS-5 or SAS-6, without which centrioles do not assemble (reviewed in [7,8]). Instead, SAS-1 exerts a function in ensuring centriole integrity. Components critical for centriole integrity have been identified in other systems, including human cells, where the inner scaffold that comprises notably POC1A/POC1B and WDR90 plays an important role [45,46]. δ- and ε-tubulin, as well as their interacting proteins TEDC1 and TEDC2, which are required for the assembly of microtubule doublets and triplets, are also important for centriole integrity in human cells [47–49]. Upon their depletion, only microtubule singlets are present, and centrioles lose structural integrity during mitosis. The ciliopathy protein HYLS1 is likewise required for doublet and triplet microtubule formation, as well as for centriole stability [50]. Given that nematode centrioles harbor 9-fold radially symmetrical microtubule singlets, instead of the usual doublets and triplets [51], instability

due to having singlet microtubules is unlikely to be the reason in *C. elegans*, although a B-like extension unveiled by cryo-electron tomography in centrioles of the embryo might contribute to stability [10]. Regardless, SAS-1 plays a critical role for centriole integrity, and it will be interesting to test whether the SAS-1 homologue C2CD3 acts in a related manner.

**SAS-1 function at sensory cilia**

We discovered a novel localization of SAS-1 at the transition zone of sensory cilia. Echoing the redundancy manifested by other components contributing to ciliary function in *C. elegans* [33,36], we found that SAS-1 contributes to ciliary function in a partially redundant manner with the transition zone protein NPHP-4. Given its apparent localization within the confines of the axonemal microtubules, we speculate that SAS-1 may be part of the central cylinder, which also comprises CCEP-290 [33]. By analogy with how SAS-1 operates in the context of centriole integrity, we further speculate that it may contribute to maintaining the ciliary axoneme in place during ciliogenesis. In human cells, C2CD3 is required for full-fledged centriole formation, as well as cilium generation, together with its upstream regulator Cep120 [52]. Therefore, in both worms and human beings, SAS-1/C2CD3 is critical for building a properly functioning sensory cilium.

**Relationship of SAS-1 with SSNA-1**

Although many proteins that reside at the centriole and the PCM have been identified in *C. elegans* through forward genetics and RNAi-based functional genomic approaches, together with others [41], we establish SSNA-1 as a novel centriolar component in the worm. In mammalian systems, SSNA1 has been described to localize to centrioles, primary cilia, mirroring the findings in worms, as well as axon branch sites [53–56]. *In vitro*, SSNA1 self-assembles into head-to-tail fibrils that have been reported to bind microtubules [57]. Likewise, *C. elegans* SSNA-1 can form such tetrameric polymer assemblies [58]. Furthermore, human SSNA1 has also been found to interact with the microtubule severing enzyme Spastin [53].

   Just like SAS-1, SSNA-1 contributes to centriole integrity in *C. elegans*, although to a lesser extent, judging from the milder phenotypes of *ssna-1*(*bs182*) null mutant animals [41; this work]. Accordingly, whereas SSNA-1 contributes to SAS-1 recruitment or maintenance at centrioles, we found that SAS-1 is essential for the presence of SSNA-1 at centrioles and cilia. Interestingly, recent work demonstrates that C2CD3 likewise recruits SSNA-1 to centrioles in human cells, mirroring the relationship between the related *C. elegans* proteins (Jen-Hsuan Wei, personal communication). We found that *C. elegans* SSNA-1 does not associate with microtubules in a heterologous human cell assay, but is recruited to the polymer upon co-expression of functional SAS-1. Taken together, these findings indicate that SAS-1 is a microtubule-binding protein that, in conjunction with SSNA-1, forms the central tube of *C. elegans* centrioles. Furthermore, C2CD3 and SSNA1 appear to exhibit an analogous relationship in human cells (Jen-Hsuan Wei, personal communication). Given that SSNA1 interacts with the microtubule severing enzyme Spastin in human cells [53], perhaps C2CD3/SAS-1 together with SSNA1/SSNA-1 shield centriolar microtubules from being severed precociously.

## Conclusion

In conclusion, our analysis with novel deletion and tissue-specific depletion alleles of SAS-1 sheds new light on the mechanisms governing integrity of the centriole organelle.

## Materials and methods

### *C. elegans* strains

Worms were grown using standard protocols on nematode growth medium (NGM) plates seeded with *Escherichia coli* OP50 as food source [59]. Strains used in this study are listed in S1 Table. Animals were typically raised at 20–22°C except for temperature-sensitive strains which were grown at 16°C until the L4 stage and then shifted to 24°C before

imaging in adulthood. Synchronized worm populations were obtained either by bleaching gravid worms and letting embryos hatch in M9 minimal medium, or by allowing 20–30 gravid adult worms to lay embryos on NGM plates for 1–2 hours and then removing the adults.

## Cell culture

U2OS cells were cultured at 37°C with 5% $CO_2$ in DMEM supplemented with GlutaMAX (ThermoFisher Scientific) and 10% fetal bovine serum. For the U2OS::SAS-1-GFP and U2OS::SAS-1(P419S)-GFP stable cell lines [23], the medium was supplemented with 1 µg/mL puromycin. To generate the U2OS::SAS-1(ΔC)-GFP stable cell line, a deletion variant lacking amino acids 546–570 of SAS-1 was produced by PCR amplification of a full-length *sas-1* plasmid [23], with primers omitting this region. The resulting linearized plasmid was sequenced and transferred into the pEBTet-GFP destination vector by LR Clonase reaction. This construct was transfected into U2OS cells, followed by 1 µg/mL puromycin selection for two weeks. For transient expression experiments, full-length *ssna-1* cDNA fused to a Myc tag was cloned into a pDONR221 vector using BP Clonase reaction (ThermoFisher Scientific). Next, *ssna-1* was transferred to a pEBTet destination vector using LR Clonase reaction, where *ssna-1* is under the control of a doxycycline-inducible CMV promoter. Cells were seeded on coverslips in a 6-well plate and allowed to grow to ~60–70% confluency before transfection, which was performed using Lipofectamine 3000 (ThermoFisher Scientific) following the manufacturer's protocol, using 2 µg of the cloned *ssna-1*-Myc plasmid. Post-transfection, cells were incubated for 4–6 hours, after which the medium was replaced with fresh medium containing 1 µg/mL doxycycline (Merck). Cells were then allowed to express the transfected construct for 48 hours, before fixation with prechilled methanol at -20°C for 7 min, followed by three washes with PBS and immunostaining.

## Gonad and sperm spreading

Gonad spreading was performed similarly to [11]. In brief, gonads from 1000-2000 young adult hermaphrodites were dissected in 30 µL of PBS (20% in $H_2O$) with Levamisole (1 mg/mL). 10 µL of this solution was put on a clean 22 x 40 mm coverslip and 50 µL spreading buffer [32 µL of fixation solution (4% w/v Paraformaldehyde and 3.2% w/v Sucrose in $H_2O$), 6 µL of Lipsol solution [1% v/v Lipsol in $H_2O$] and 2 µL of Sarcosyl solution (1% w/v of Sarcosyl in $H_2O$)] was added. The coverslips were dried for 1 hour at room temperature and a further 3 hours at 37°C. Thereafter, coverslips were either stored at -80°C or processed further for immunostaining or expansion microscopy. For sperm spreading, ~1000 hermaphrodites or males were washed and resuspended in 100 µL 1x nuclear purification buffer (10 mM HEPES, 40 mM NaCl, 90 mM KCl, 2 mM EDTA, 0.5 mM EGTA, 0.2 mM DTT, 0.5 mM Spermidine, 0.1% Triton-X). The worms were then homogenized by twenty strokes with a tight pestle with a 90° turn at each stroke, before spinning down 30 µL of this suspension onto 8 mm round coverslips (10,000g for 5 min). The coverslips were removed and the specimen fixed with prechilled methanol at -20°C for 5 min, washed further with PBST and processed for expansion microscopy.

## Gonad and sperm expansion for U-Ex-STED

Ultrastructure expansion microscopy coupled to STED was performed based on [11]. Briefly, dried coverslips containing spread gonads were further fixed with prechilled methanol at -20°C for 20 min, followed by three 10 min washes with PBS-T (this step was skipped for spread sperm nuclei). The coverslips were then incubated overnight at room temperature in Acrylamide/Formaldehyde solution (1%/0.7% in PBS). On the following day, the coverslips underwent three 5 min washes with PBS. For the gelation step, 22 x 40 mm coverslip were incubated with ~250 µL (20 µL for 8 mm coverslips) of monomer solution (19% (wt/wt) Sodium Acrylate, 10% (wt/wt) Acrylamide, 0.05% (wt/wt) BIS in PBS), supplemented with 0.5% Tetramethylethylenediamine (TEMED) and 0.5% Ammonium Persulfate (APS) for 1 hour in a humid chamber protected from light. The resulting gels were then incubated with denaturation buffer (200 mM SDS, 200 mM NaCl, and 50 mM Tris (pH 9)) for 1 hour at 70°C. Post-denaturation, gels were washed five times with distilled water for 20 min each,

followed by an overnight incubation in distilled water. The expansion factor was calculated by measuring the dimensions of the expanded gels with a ruler and comparing them with those of the coverslips. For immunostaining, the gels were first incubated in blocking buffer (10 mM HEPES (pH 7.4), 3% BSA, 0.1% TWEEN 20, 0.05% sodium azide) for 1 hour at room temperature. They were then incubated overnight with primary antibodies diluted in blocking buffer at room temperature. The next day, after three 10 min washes with blocking buffer, gels were incubated with secondary antibodies and 0.7 μg/mL Hoechst 33258 for 3 hours at 37°C. Post-incubation, gels were re-expanded through three 10 min incubations in distilled water. Finally, gels were mounted between two 22 x 60 mm coverslips coated with poly-D-lysine (2 mg/mL in water). The mounted gels were sealed using VaLaP, a 1:1:1 mixture of petroleum jelly, lanolin, and paraffin wax.

## Immunostainings

L1 larvae immunostaining was performed using a modified version of the protocol described in [27]. 6 μL of age-synchronized larvae suspension in water was put on Poly-D-Lysine (2 mg/mL in water) coated frosted slides (Marienfeld, 1000200). An 18 x 18 mm coverslip was placed gently on the larvae and excess water removed with a blotting paper. The slide was quickly placed on a prechilled metal block kept on dry ice for a minimum of 10 min. Thereafter, the coverslip was flicked using a scalpel or razor blade and the slide dipped in prechilled methanol at -20°C for 2 min for fixation. Slides were then washed 3 times in PBST to remove traces of methanol, followed by blocking for 30 min in 2% BSA in PBST. Primary antibodies in blocking buffer were incubated either for 1–2 hours at room temperature or overnight at 4°C. Slides were then washed three times in PBST for 10 min each. Secondary antibodies in PBST were incubated at room temperature for 45 min, followed by washing and 0.7 μg/L Hoechst 33258 staining in PBST. The slides were washed twice before mounting in 4% n-Propyl-Gallate, 90% Glycerol, 1x PBS, and coverslips sealed with colorless nail polish.

Immunostaining on whole dissected gonads was performed based on [21]. Gonads from young adult hermaphrodites were dissected in sperm buffer (50 mM HEPES [pH 7.0], 50 mM NaCl, 25 mM KCl, 5 mM CaCl$_2$, 1 mM MgSO$_4$, 50 mM Glucose, 1 mg/mL BSA) with Levamisole (1mg/mL), and gently placed on frosted glass slides coated with Poly-D-Lysine (2 mg/mL in PBS), followed by lowering of the coverslip (22 x 40 mm) and freezing on dry ice. The coverslip was flicked as above and the slide dipped in prechilled methanol at -20°C for 5 min. The remaining steps, including washing, blocking and antibody incubation, were similar to those mentioned for immunostaining of larvae.

For immunostaining of U2OS cells, fixed cells were blocked for 30 min using 3% BSA in PBS with 0.05%(v/v) TWEEN 20 (PBST), followed by primary antibody incubation in the blocking buffer for 1 hour at room temperature. This was followed by three washes with PBST, and secondary antibody incubation with 0.7 μg/L Hoechst 33258 for 1 hour at room temperature. Cells were washed again and mounted on slides using Fluoromount G (Invitrogen).

Primary antibodies used in this study: rabbit anti-SAS-4: 1:5000 (immunostaining or U-Ex-STED) [60]; rabbit anti-GFP: 1:1000 (immunostaining) (gift from Viesturs Simanis); rabbit anti-α-tubulin 1:5000 (Western blotting) (Abcam | ab52866); mouse anti-FLAG: 1:500 (U-Ex-STED) (Thermo | MA1–91878); mouse anti-FLAG: 1:200 (immunostaining), 1:3000 (Western blotting) (Merck | F1804); SPOT: 1:500 (U-Ex-STED) (Chromotek | 28a5-20); mouse anti-α/β-tubulin: 1:500 (immunostaining and U-Ex-STED) [61]; mouse anti-Myc 1:500 (immunostaining) (CST | Ab#2276S); SPOT nanobody Alexa Fluor 568: 1:1000 (immunostaining) (Chromotek | ebAF568); SPOT nanobody Alexa Fluor 488: 1:1000 (immunostaining) (Chromotek | ebAF488).

The following secondary antibodies were used at 1:1000 (immunostaining) or 1:500 (U-Ex-STED): goat anti-mouse Alexa Fluor 488; goat anti-rabbit Alexa Fluor 568; donkey anti-rabbit Alexa Fluor 594; goat anti-mouse Alexa Fluor 647.

## Microscopy

Four confocal microscopy systems were utilized. First, an upright Leica SP8 equipped with two hybrid photon counting detectors (HyD) and a transmission photomultiplier tube (PMT) for brightfield imaging. This system employs a 63x HC Plan-Apochromat objective (NA 1.4) and 405, 488, and 552 nm solid-state laser lines for excitation, paired with a DFC

7000 GT (B/W) camera. Second, an inverted Olympus IX83 motorized microscope equipped with Yokogawa spinning disk CSU-W1 head, a 60x (NA 1.42 U PLAN S APO) objective, as well as ImagEMX2 EMCCD and Orca Flash 4.0 sCMOS cameras, with image acquisition controlled by VisiView software. Third, 2D-STED (Stimulated Emission Depletion) images were captured on a Leica TCS SP8 STED 3X microscope, using a 100x 1.4 NA oil-immersion objective. The system employs 488 and 589 nm excitation lasers, along with 592 and 775 nm pulsed lasers for depletion. Finally, Airyscan imaging was performed using an inverted Zeiss LSM 980 microscope equipped with Airyscan 2 detection, featuring a 63× (NA 1.4) oil-immersion Plan-Apochromat objective and a 32-channel Airyscan 2 area detector.

Live imaging of late-stage embryos (Fig 3A) was performed using a Zeiss Lattice Lightsheet 7 microscope equipped with a 44.83×/1.0 NA water-immersion objective. Embryos were maintained at 22–24 °C during imaging. Volumetric image stacks were acquired every 10 minutes using 488 and 561 nm solid-state lasers for excitation; images were captured on a PCO Edge 4.2 sCMOS grayscale camera.

Live imaging of early embryos, along with cilium and dendrite length measurement, were performed on a Zeiss Axio Observer D.1 inverted microscope equipped with a 63x NA 1.4 Plan-Apochromat objective, connected to an Andor Zyla 4.2 sCMOS camera and an LED light source (Lumencor SOLA II).

## Image processing and analysis

Visualization and quantification of microscopy images were performed using Fiji (ImageJ) [62]. For quantification of nuclear area in Fig 1E, the Z-plane containing the largest cross-sectional area of each nucleus was identified. An ellipse was visually fitted to this plane, its area measured and used as a proxy for nuclear size.

For presentation/quantification of 2D-STED images (Figs 1G–1I and 2B–2E), individual channels were centered to correct for chromatic shifts. A 1-pixel Gaussian blur was applied to all images for both analysis and display. For display purposes only, to account for variable zoom factors across image sets, regions of interest (ROIs) of identical pixel dimensions were drawn on each image and then uniformly scaled to match the smallest field of view. Brightness and contrast were individually adjusted for each image. To measure centriole diameter, a line was drawn across each centriole, and a super-Gaussian curve fitted to the intensity profile. The full width at half maximum (FWHM) of the fit was taken as the centriole diameter.

For Fig 3B, GFP::SAS-1 foci were quantified from Lattice lightsheet microscopy datasets using a custom Fiji macro. A user-defined ROI was drawn to restrict analysis to the embryo region, and a Laplacian of Gaussian (LoG) detector was applied frame-by-frame to identify potential centrioles. Spot detection was performed on a per-channel and per-timepoint basis, using subpixel localization and intensity thresholds optimized for each channel. Only foci within the user-specified ROI were retained, and their positions and intensities were saved for further analysis.

For signal intensity quantification in Figs 5G and 6C, sum-projected images of gonads were divided into eight (Fig 5G) or seven (Fig 6C) regions defined by ~3 × the diameter of diplotene nuclei, measured at the onset of cellularization (Fig 5G) or at the gonad loop (Fig 6C). ROIs of 1.8 $\mu m^2$ were placed on each focus and an adjacent background area. Background-corrected intensities were calculated per channel and normalized to the most distal region. For S2D–S2I Fig, 15 × 15-pixel ROIs were centered on each focus and a nearby background region at the z-slice of maximal intensity. A 6.3 µm sum projection centered on this slice was generated, and background-subtracted signal was measured.

## Ciliary assays

For cilia staining in Fig 4E (L1-L2 larvae) and Fig 4G–4J (L4 larvae), animals were washed in M9 buffer and incubated with 1 µM SPY650-tubulin (Spirochrome) for 3 hours. After incubation, worms were washed again in M9 buffer and immediately imaged. For the dye-filling assay in Fig 4L and S5A–S5D Fig, about 200 L4 larvae were washed off from NGM plates in M9 and incubated with 10 µg/mL of DiI for 3 hours. Worms were allowed to destain for 30 min on NGM plates before scoring/imaging.

The chemotaxis assay was conducted based on [40]. Approximately 50–100 well-fed adult worms were collected from NGM plates and placed in an Eppendorf tube with M9 buffer, and washed twice with M9 to remove any residual bacteria. A 10 cm NGM assay plate was prepared by marking two points in a straight line along its diameter and located near the periphery of the plate. At one of these points, 1 μL of 100% isoamyl alcohol (attractant) mixed with 1 μL 10% sodium azide was applied. At the opposite point, 1 μL of 100% ethanol (repellent) mixed with 1 μL 10% sodium azide was added. The sodium azide served to paralyze the worms, facilitating accurate counting. The washed worms were then transferred to the center of the plate, approximately 5 cm from both attractant and repellent point sources. After 1 hour, the distribution of worms was analyzed. Worms that had migrated toward either end - i.e., within approximately a 2.5 cm radius of the attractant or repellent point were counted and analyzed.

The chemotaxis index was calculated using the following formula:

$$\text{Chemotaxis index} = \frac{\text{No. of worms at attractant point} - \text{No. of worms at repellent point}}{\text{Total number of worms}}$$

The chemotaxis index ranged from -1 (perfect repellent) to 1 (perfect attractant).

## Supporting information

**S1 Fig. Generation of *sas-1* null allele. A.** Schematic of wild-type *sas-1*, as well as *sas-1(is6)* [11], and *sas-1(is13)* alleles. Primers at the N- and C-terminus used for PCR-based genotyping are indicated. **B.** Agarose gel image of PCR products from genomic DNA of control and *sas-1(is13)* mutant worms using primers mentioned in (A). **C.** Sanger sequencing showing 74 bp deletion in *sas-1(is13)*, stretching from the 5'-UTR to Exon 1, leading to loss of the ATG start codon. **D.** Western blotting of worm lysates. Note that the ~65 kDa band of *sas-1::3xflag* is lost in *sas-1(is13)*. **E.** Progeny test of *sas-1(is13)/hT2(gfp)* worms; N = 8 technical repeats. **F.** Schematic of genotypes among progeny derived from strain *tagRFP-T::sas-1/sas-1(is13)* with indication of corresponding animals in S1G and S1H Fig (Created and adapted from BioRender (https://BioRender.com/s7jx4jh)). **G, H.** Representative live images of gonads of worms expressing GFP::SAS-7 and possibly zygotic or/and maternal TagRFP-T::SAS-1 (G, N = 8 gonads), as well as GFP::SAS-7 and maternal TagRFP-T::SAS-1 only (H, N = 6 gonads).
(TIF)

**S2 Fig. Level of SAS-7 and SAS-4 in *sas-1(is13)*. A, B.** Representative immunofluorescence images of gonads of control (A) and *sas-1(is13)* mutant (B), both expressing GFP::SAS-7 and immunostained for GFP and SAS-4. Numbered regions in white mark the positions where signal quantification, shown in C, was performed. **C.** Quantification of percentage of nuclei with or without centrioles in different regions of distal gonad of control (N = 6 gonads) and *sas-1(is13)* mutants (5 gonads; N = 230 nuclei for region 1; N = 236 nuclei for region 2; N = 208 nuclei for region 3). **D, E, G, H.** Live imaging of mitotic region of the gonad in control (D, G) or *sas-1(is13)* mutant (E, H) worms expressing SAS-4::GFP (D, E) or GFP::SAS-7 (G, H). Three regions of ~18 μm in width and ~18 μm apart were defined as indicated. Signal intensity of SAS-4::GFP and GFP::SAS-7 was then quantified in these regions. **F, I.** Quantification of signal intensity of SAS-4::GFP (F) and GFP::SAS-7 (I) in the three regions described in (D, E, G, H). Foci number for F: control (5 gonads; N = 58 for region 1, 61 for region 2, 61 for region 3), *sas-1(is13)* (5 gonads; N = 48 for region 1, 60 for region 2, 54 for region 3). Foci number for I: control (5 gonads; N = 58 for region 1, 55 for region 2, 59 for region 3), *sas-1(is13)* (5 gonads, N = 56 for region 1, 56 for region 2, 51 for region 3). Student's two-tailed t-tests, whereby $P < 0.001$ (***). The differences between control and *sas-1(is13)* in F are not significant (Region 1: $P = 0.77$, Region 2: $P = 0.055$, Region 3: $P = 0.69$).
(TIF)

**S3 Fig. Monitoring centrioles in control *and sas-1(is13)* female germline. A.** Representative U-Ex-STED images of centrioles immunostained for SAS-4 in the mitotic zone, early prophase, and late prophase of control (N = 15, 14 and

20, respectively) and *sas-1(is13)* mutant (N = 18, 10 and 10, respectively) gonads. Scale bar is corrected for the expansion factor of 5. B. Quantification of centriole width in indicated regions of control and *sas-1(is13)* mutant gonads from A. Student's two-tailed t-tests, whereby $P < 0.05$ (*); $P < 0.01$ (**); $P < 0.001$ (***). C, D. *sas-1(is13)* mutant hermaphrodites expressing SAS-4::GFP (green) mated with control males expressing TagRFP-T::SAS-7 (magenta). (Created and adapted from BioRender (https://BioRender.com/s7jx4jh) as well as [63]). E. Possible outcomes of marked mating experiment: (left) Control; in this case, in cycle 2, two SAS-4::GFP foci are present in both blastomeres, which undergo bipolar spindle assembly. (middle) SAS-1 is essential for centriole assembly; in this case, no SAS-4::GFP foci are present in the embryo, and monopolar spindle assembly occurs in both blastomeres at the two-cell stage. (right) SAS-1 is essential for centriole integrity; in this case, two SAS-4::GFP foci are present in each blastomere initially but are eliminated thereafter (indicated by black cross). F, G. Snapshots from live imaging at the time of pronuclear meeting in control (F; N = 10) and *sas-1(is13)* mutant (G; N = 9) one-cell stage embryos. Images were acquired every minute. The DIC channel is a single plane, whereas the GFP::SAS-7 channel is a maximum intensity projection of selected Z-planes. Whereas both sperm-contributed centrioles are invariably visible at pronuclear meeting in the control (N = 10), no focus of GFP::SAS-7 is present at the equivalent stage in *sas-1(is13)* mutant embryos (N = 9), although one was detected at earlier stages in some embryos (3/9).
(TIF)

**S4 Fig. SAS-1 plays a role in cilia function. A.** Live imaging of sensory cilia at anterior of animals expressing TagRFP-T::SAS-1 and GFP::SPD-5 during larval stages and adulthood, as indicated. N = 10 for each stage. The bright cylindrical signal observed in the TagRFP-T channel in the adult stage is likely autofluorescence from the mouth orifice. **B.** Schematic of ZIF-1 mediated degron system, with ZIF-1 expression under the control of the *dyf-7* promoter in this case. **C-F.** Live imaging of L1 larvae expressing TagRFP-T::SPD-5 and ZF::GFP::SAS-1 in control (C), *nphp-4(tm925)* (D), *Pdyf-7::degron* (E, note two remaining ZF::GFP::SAS-1 foci, likely due to lack of ZIF-1 expression in these neurons), and *Pdyf-7::degron nphp-4(tm925)* (F) animals. **G.** Quantification of TagRFP-T::SPD-5 intensity in S4C–S4F Fig. Rectangular ROI of 15.82 μm² was drawn around amphid cilia or adjacent worm body (for background subtraction) in the sum projected Z-slices of 6 μm. TagRFP-T::SPD-5 signal reported is the background subtracted mean signal intensity of amphid cilia. N = 26 (control, from 13 animals); N = 24 (*nphp-4(tm925)*, from 12 animals); N = 22 (*Pdyf-7::degron*, from 11 animals), N = 22 (*Pdyf-7::degron nphp-4(tm925)*, from 11 animals). Student's two-tailed t-tests, which were all not significant compared to control; *nphp-4(tm925)*: $P = 0.34$, *Pdyf-7::degron*: $P = 0.07$, *Pdyf-7::degron nphp-4(tm925),* $P = 0.05$. **H-K.** Live imaging of L1 larvae expressing CHE-11::mKate2 and ZF::GFP::SAS-1 in control (H), *nphp-4(tm925)* (I), *Pdyf-7::degron* (J), and *Pdyf-7::degron nphp-4(tm925)* (K) animals. **L.** Quantification of CHE-11::mKate2 intensity in S4H–S4K Fig. Quantification was performed as mentioned in S4G Fig. N = 12 (control, from 6 animals); N = 12 (*nphp-4(tm925)*, from 6 animals); N = 12 (*Pdyf-7::degron*, from 6 animals), N = 15 (*Pdyf-7::degron nphp-4(tm925)*, from 8 animals). Student's two-tailed t-tests and comparison was made with control; *nphp-4(tm925)*: $P < 0.001$ (***), *Pdyf-7::degron*: $P = 0.16$, *Pdyf-7::degron nphp-4(tm925),* $P < 0.001$ (***). Comparison between nphp-4(tm925) vs *Pdyf-7::degron nphp-4(tm925)*: $P = 0.06$.
(TIF)

**S5 Fig. Dendrite length measurement and chemosensation assay. A-D.** Representative image of phasmid neurons stained with DiI in control (A), *nphp-4(tm925)* (B), *Pdyf-7::degron* (C), *Pdyf-7::degron nphp-4(tm925)* (D) animals. Dendrite length was measured as illustrated by drawing a segmented line from the end of the neuron cell body until the beginning of cilium (which stains much more faintly than rest of the neuron by DiI and can hence be identified easily). Images are maximum intensity projections of selected Z-planes. **E.** Quantification of dendrite length from A-D. Dendrites quantified: N = 27 (control, from 14 animals); N = 27 (*nphp-4(tm925)*, from 14 animals); N = 26 (*Pdyf-7::degron*, from 14 animals), N = 23 (*Pdyf-7::degron nphp-4(tm925)*, from 12 animals). Student's two-tailed t-tests, which were all not significant compared to control vs *nphp-4(tm925)*: $P = 0.06$, *Pdyf-7::degron*: $P = 0.4$, *Pdyf-7::degron nphp-4(tm925),* $P = 0.09$. **F.** Schematic

of chemosensation assay (not to scale) (Created and adapted from BioRender (https://BioRender.com/8ba9yul)). See text for further details.
(TIF)

**S6 Fig. Interaction between SAS-1 and SSNA-1. A-D.** Snapshots from time-lapse imaging showing DIC and GFP::SAS-7 distribution in embryos at pronuclear meeting in control (A; N = 10), *sas-1(t1476)* (B; N = 10), *ssna-1(bs182)* (C, N = 10), and *sas-1(t1476); ssna-1(bs182)* (D; N = 7) embryos. DIC channel is a single plane whereas the GFP::SAS-7 channel is a maximum intensity projection of selected Z-planes. **E.** AlphaFold2 prediction of SAS-1 with two C2-domains and a C-terminal alpha helix. The flexible loop connecting the two C2 domains is indicated with dotted line. **F.** AlphaFold2 prediction of SAS-1 binding to SSNA-1 tetramer. SSNA-1 is forming a head-to-tail fibril with the C-terminal helix of SAS-1 binding to the interface of two helixes. Top-right: magnified representation of the interface marked with dotted line; SAS-1 in cyan, SSNA-1 in pink.
(TIF)

**S7 Fig. SAS-1 contributes to SSNA-1 distribution at sensory cilia. A.** Live imaging of anterior cilia of L1 larvae expressing TagRFP-T::SAS-1 in control (N = 12) and *ssna-1(bs182)* mutant (N = 13). **B.** Quantification of TagRFP-T::SAS-1 signal intensity in anterior sensory cilia from S7A. Quantification was performed as mentioned in S4G Fig with a ROI of ~12 $\mu m^2$. N = 24 from 12 control worms, and N = 26 from 13 *ssna-1(bs182)* mutants. Student's two-tailed t-tests, whereby $P < 0.05$ (*). **C.** Representative immunofluorescence image of L1 larva stained for ZF:GFP::SAS-1 (green) and SSNA-1::SPOT (magenta). Insets are 2 times magnified.
(TIF)

**S8 Fig. Schematic of potential impact of SAS-1 loss on centriole integrity. A, B.** Schematic of two rounds of centriole duplication in the mitotic region of the female germline in control (A) and *sas-1(is13)* mutant (B) animals. Dashed lines indicate cytoplasmic area around individual germ line nuclei. In the control, the two pre-existing centrioles ($C_0$, magenta) each gave rise to one new centriole ($C_1$, green), which matured during mitosis (top row). In the following interphase, all four centrioles (2x $C_0$ and 2x $C_1$) mentor assembly of one new centriole each ($C_2$, blue) (second row). In the following mitosis, $C_2$ centrioles mature (third row). In the next interphase (last row), all centrioles mentor assembly of one new centriole each (white). In *sas-1(is13)* mutants, centrioles are generated, but lose structural integrity as they traverse the subsequent mitoses. As a result, germ cell nuclei at the end of the second mitosis exhibit two types of configurations. In both types, the first, younger, centriole ($C_2$) is intact, whereas the second, older, centriole is either deformed ($C_1$) or absent ($C_0$), depending whether it was formed in the previous cell cycle ($C_1$) or earlier than that ($C_0$). Note also that ~20% of *sas-1(is13)* nuclei are devoid of centrioles (see S2C Fig), so that organelle loss must be more drastic in some cases, perhaps in those configurations with the oldest $C_0$ centrioles. **C, D.** Schematic of centriole duplication during the meiotic divisions of spermatogenesis in control (C) and *sas-1(is13)* mutant (D) animals. In the control, the two pre-existing centrioles ($C_0$, magenta) each gave rise to one new centriole prior to the first meiotic division ($C_1$, green). All four centrioles (2x $C_0$ and 2x $C_1$) mentor assembly of one new centriole each ($C_2$, blue) prior to the second meiotic division. In *sas-1(is13)* mutants, centrioles are assembled as in the control, but the two that were present initially ($C_0$) lose structural integrity during the two meiotic divisions, so that they are ruptured in mature sperm. As a consequence, sperm exhibit two configurations types: one in which both centrioles appear intact, albeit widened, and another one in which one of the centrioles is ruptured.
(TIF)

**S1 Movie. Lattice light sheet imaging of embryos expressing TagRFP-T::SAS-1 and GFP::SAS-7.** Embryos expressing TagRFP-T::SAS-1 (magenta) and GFP::SAS-7 (green) were imaged using lattice light sheet microscopy. Images were acquired every 10 min. Each frame is the maximum intensity projection of all slices containing the embryos. The time stamp (upper left) indicates minutes post-fertilization.
(AVI)

**S1 Table. List of *C. elegans* strains used in this study.** The table provides the complete list of *C. elegans* strains used in this study.
(PDF)

**S1 Dataset. Data underlying all plots presented in this study.** This dataset contains individual sheets with the data used to generate the plots included in this work. It also includes the corresponding statistical analyses and summary values (e.g. mean, SD, etc.).
(XLSX)

## Acknowledgments

We thank Kevin O'Connell and Jason Pfister for sharing *ssna-1*(*bs182*) *and ssna-1::spot* worm strains prior to publication, as well as for fruitful discussions, Jen-Hsuan Wei for communicating results prior to publication, Cédric Pourroy for help with the SPY650-tubulin ciliary marking experiment, Alana Dastous for help with AlphaFold multimer, Rémi Dornier (BioImaging and Optics Platform of the School of Life Sciences, EPFL) for a macro to quantify GFP::SAS-1 foci. We are grateful to Friso Douma, Gabriela Garcia-Rodriguez and Marie Pierron for constructive comments on the manuscript.

## Author contributions

**Conceptualization:** Keshav Jha, Alexander Woglar, Pierre Gönczy.

**Data curation:** Keshav Jha, Pierre Gönczy.

**Formal analysis:** Keshav Jha, Pierre Gönczy.

**Investigation:** Keshav Jha, Alexander Woglar, Coralie Busso, Georgios N. Hatzopoulos, Tatiana Favez, Pierre Gönczy.

**Methodology:** Keshav Jha, Pierre Gönczy.

**Supervision:** Alexander Woglar, Pierre Gönczy.

**Writing – original draft:** Keshav Jha, Pierre Gönczy.

**Writing – review & editing:** Keshav Jha, Alexander Woglar, Pierre Gönczy.

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
