## [Editor Report · Decision Letter 0]

16 Jun 2025

PGENETICS-D-25-00687

C. elegans SAS-1 ensures centriole integrity and ciliary function, and operates with SSNA-1

PLOS Genetics

Dear Dr. Gonczy,

Thank you for tranfering your manuscript from Reviewers commons to PLOS Genetics. We would be pleased to receive a revised version of your manuscript based on your detailed plan to addresses the points raised during the review process.

Please submit your revised manuscript within 60 days Aug 15 2025 11:59PM. If you will need more time than this to complete your revisions, please reply to this message or contact the journal office at plosgenetics@plos.org. Please include the following items when submitting your revised manuscript:

We look forward to receiving your revised manuscript.

Kind regards,

Nathalie Pujol

Academic Editor

PLOS Genetics

Pablo Wappner

Section Editor

PLOS Genetics

Aimée Dudley

Editor-in-Chief

PLOS Genetics

Anne Goriely

Editor-in-Chief

PLOS Genetics

**Journal Requirements:**

https://journals.plos.org/plosgenetics/s/submission-guidelines#loc-parts-of-a-submission

4) We notice that your supplementary Figures, and Table are included in the manuscript file. Please remove them and upload them with the file type 'Supporting Information'. Please ensure that each Supporting Information file has a legend listed in the manuscript after the references list.

Potential Copyright Issues:

i) Figures 4K, and S3. Please confirm whether you drew the images / clip-art within the figure panels by hand. If you did not draw the images, please provide (a) a link to the source of the images or icons and their license / terms of use; or (b) written permission from the copyright holder to publish the images or icons under our CC BY 4.0 license. Alternatively, you may replace the images with open source alternatives. See these open source resources you may use to replace images / clip-art:

ii) Figures 2F, and S4D. We note that the figures are created through BioRender. Please confirm that you hold a Premium account and provide a pdf copy of the CC BY 4.0 Licence as provided by BioRender. For instructions on how to generate a CC BY 4.0 license for your figure, please see the guidelines here: https://help.biorender.com/hc/en-gb/articles/21282341238045-Publishing-in-open-access-resources. 

If you are using the free assets from BioRender, we are unable to publish these images as they are licenced under a stricter licence than CC BY 4.0. In this case we ask you to remove the BioRender images and replace them with open source alternatives.

See these open source resources you may use to replace images / clip-art:

- https://bioart.niaid.nih.gov/ 

- https://bioicons.com/

- https://healthicons.org/ 

- https://scidraw.io/

- https://reactome.org/icon-lib

- https://www.phylopic.org/images 

- https://journals.plos.org/plosbiology/article?id=10.1371/journal.pbio.3002395

6) We note that your Data Availability Statement is currently as follows: "All relevant data are within the manuscript and its Supporting Information files." Please confirm at this time whether or not your submission contains all raw data required to replicate the results of your study. Authors must share the “minimal data set” for their submission. PLOS defines the minimal data set to consist of the data required to replicate all study findings reported in the article, as well as related metadata and methods (https://journals.plos.org/plosone/s/data-availability#loc-minimal-data-set-definition).

7) Please amend your detailed Financial Disclosure statement. This is published with the article. It must therefore be completed in full sentences and contain the exact wording you wish to be published.

**Figure resubmission:**
---

## [Decision Letter · Decision Letter 1]

8 Oct 2025

Dear Dr Pierre Gönczy,

We are pleased to inform you that your manuscript entitled "C. elegans SAS-1 ensures centriole integrity and ciliary function, and operates with SSNA-1" has been editorially accepted for publication in PLOS Genetics. Congratulations!

Yours sincerely,

Nathalie Pujol

Academic Editor

PLOS Genetics

Pablo Wappner

Section Editor

PLOS Genetics

Aimée Dudley

Editor-in-Chief

PLOS Genetics

Anne Goriely

Editor-in-Chief

PLOS Genetics

BlueSky: @plos.bsky.social

Comments from the reviewers (if applicable):

Reviewer's Responses to Questions

**Comments to the Authors:**

Reviewer #1: The authors have addressed my concerns. I am in favor of publication.

Reviewer #2: The authors have addressed all of my comments. In particular, clarification about the phenotype of the sas-1 null mutant and additional information on the role of SAS-1 in cilia have improved and strengthened the paper. I recommend the manuscript for publication.

Reviewer #3: I appreciate the authors' thorough responses and the results from the new experiments, which satisfactorily addressed my questions and concerns. I now recommend this article for publication in PLOS Genetics.

**Have all data underlying the figures and results presented in the manuscript been provided?**

Reviewer #1: Yes

Reviewer #2: None

Reviewer #3: Yes

PLOS authors have the option to publish the peer review history of their article (what does this mean? ). If published, this will include your full peer review and any attached files.

**Do you want your identity to be public for this peer review?** For information about this choice, including consent withdrawal, please see our Privacy Policy .

Reviewer #1: No

Reviewer #2: No

Reviewer #3: No

**Data Deposition**

http://datadryad.org/submit?journalID=pgenetics&manu=PGENETICS-D-25-00687R1

**Press Queries**

---

## [Editor Report · Acceptance letter]

PGENETICS-D-25-00687R1

C. elegans SAS-1 ensures centriole integrity and ciliary function, and operates with SSNA-1

Dear Dr Gönczy,

We are pleased to inform you that your manuscript entitled "C. elegans SAS-1 ensures centriole integrity and ciliary function, and operates with SSNA-1" has been formally accepted for publication in PLOS Genetics! Your manuscript is now with our production department and you will be notified of the publication date in due course.

With kind regards,

Olena Szabo

PLOS Genetics

On behalf of:
